**Subject Category:**
Biology (whole organism)

evolution/palaeontology

facultative bipedality, locomotor evolution, evolutionary transitions, ancestral state reconstruction, palaeontology, macroevolution

**Author for correspondence:**
Luke R. Grinham
e-mail: lg515@cam.ac.uk

# Testing for a facultative locomotor mode in the acquisition of archosaur bipedality

Luke R. Grinham[1], Collin S. VanBuren[2]
and David B. Norman[1]

[1]Department of Earth Science, University of Cambridge, Downing Street, Cambridge CB2 3EQ, UK
[2]Department of Evolution, Ecology, and Organismal Biology, The Ohio State University, Columbus, OH 43210, USA

 LRG, 0000-0001-5583-8052; CSVB, 0000-0002-4555-3341;
DBN, 0000-0003-0101-428X

Bipedal locomotion is a defining characteristic of humans and birds and has a profound effect on how these groups interact with their environment. Results from extensive hominin research indicate that there exists an intermediate stage in hominin evolution—facultative bipedality—between obligate quadrupedality and obligate bipedality that uses both forms of locomotion. It is assumed that archosaur locomotor evolution followed this sequence of functional and hence character-state evolution. However, this assumption has never been tested in a broad phylogenetic context. We test whether facultative bipedality is a transitionary state of locomotor mode evolution in the most recent early archosaur phylogenies using maximum-likelihood ancestral state reconstructions for the first time. Across a total of seven independent transitions from quadrupedality to a state of obligate bipedality, we find that facultative bipedality exists as an intermediary mode only once, despite being acquired a total of 14 times. We also report more independent acquisitions of obligate bipedality in archosaurs than previously hypothesized, suggesting that locomotor mode is more evolutionarily fluid than expected and more readily experimented with in these reptiles.

## 1. Background

Bipedal locomotion is one of the defining characteristics of humans and birds—some of the most widely distributed vertebrate species alive today—as well as many ricochetal mammals. There are various hypotheses that attempt to explain why bipedal

locomotion is evolutionarily advantageous. In humans, for example, it has been proposed that a shift toward savannah-like aridity encouraged tree-dwelling populations of hominins on to the ground, where bipedal locomotion was demonstrably more energetically efficient for moving between increasingly distant arboreal habitats [1,2]. The most obvious advantage of using only hindlimbs to locomote is the freeing of forelimbs for use in functions other than those associated with support and locomotion. These functions can be broadly categorized as social use (communication, combat), micro-mechanical use (tool and object manipulation) and macro-mechanical use (flight, or environmental manipulation such as digging). The enhanced capacity to interact with other organisms and the surrounding environment undoubtedly contributes to the success of modern humans and birds relative to other vertebrate groups [3]. However, there is a limited understanding of how bipedality evolved in non-human and non-avian animals.

Hominin bipedality is suspected to have evolved first around 4.4 Ma with *Ardipithecus ramidus* [4], though there are some indicators of potential bipedal capability as far back as 7 Ma in *Sahelanthropus* [5]. It is widely accepted that the achievement of bipedality was not the consequence of a single event, but rather represented a progressive acquisition of anatomical features that enabled an upright posture and two-legged locomotion [6,7].

Species exhibiting a tendency to employ both bipedal and quadrupedal locomotor modes are referred to herein as facultative bipeds. Among reptiles, this locomotor mode is seen today in modern squamates, such as basilisk lizards or frilled lizards [8]. It can be argued that the facultative locomotor mode exists in two states: facultative bipedality and facultative quadrupedality, depending upon the predominant style of movement based on behavioural observation. This is a matter of evolutionary polarity: facultative quadrupedality is commonly used when describing secondarily quadrupedal animals having evolved from obligately bipedal ancestors; this contrasts with bipedal animals evolving from ancestral quadrupeds, the case that we are investigating in this study.

For birds, the origin of bipedal locomotion is rooted much deeper in their evolutionary history. Bipedality is plesiomorphic for birds, as it is for all dinosaurs [9], and its evolutionary origin is currently hypothesized to lie within dinosauromorph archosaurs [10]. In 2012, Kubo & Kubo [11] proposed that bipedality arose up to six times within archosaurs, by correlating limb proportions indicative of cursoriality with bipedalism. In 2017, Persons & Currie [10] re-iterated the hypothesis that facultative bipedality represented a transitional stage in the acquisition of bipedality in dinosauromorphs (as in hominins), although no quantitative evidence was offered. The latter authors predicted that taxa interpreted as obligate bipeds (e.g. the first dinosaurs) should have ancestors that are facultative bipeds. However, no large-scale taxon-level assessment of locomotor mode across Archosauria and their direct ancestors and descendants (Archosauriformes and Dinosauria) has been attempted within a phylogenetic framework, making it difficult to assess the validity of this prediction. Examining the sequence of character evolution across clades provides a framework to test the robustness of adaptive evolutionary hypotheses in the fossil record [12].

Here, we test the sequence in which locomotor states evolved across the transition between quadrupedal and bipedal locomotor modes using two recently published phylogenies focused on the relationships of early archosaurs and their ancestors.

## 2. Material and methods

Two recent phylogenies of early archosaurs have yielded insights into patterns of morphological evolution in this clade [13], generated from two independent character matrices created by Ezcurra [14] and Nesbitt [15]. The terms 'Ezcurra tree' and 'Nesbitt tree' will be used in this article. We used the strict consensus trees from the authors' analyses that were derived from four most parsimonious trees in the case of the Ezcurra analysis, and 36 most parsimonious trees for the Nesbitt analysis. The Ezcurra tree comprises mostly early archosauriforms, their proximate ancestors and descendants, ranging from the earliest known Carboniferous diapsid *Petrolacosaurus* through to early herrerasaurids of the Upper Triassic, with a notably large representation of Lower Triassic taxa. The Nesbitt tree focuses greater attention upon Upper Triassic archosaurs and their immediate descendants (including early dinosaurs and crocodylomorphs). The phylogenies include 107 and 83 taxa, respectively.

To determine whether each taxon was classified as an obligate quadruped (OQ), facultative biped (FB) or obligate biped (OB), we conducted a literature survey of all taxa included in the two matrices and recorded the most recent interpretation of locomotor mode for each taxon, along with the evidence thereof (electronic supplementary material, file S1). The methods used by authors to determine locomotor mode varied considerably. Taxa diagnosed as primarily or semi-aquatic were

classified as obligate quadrupeds, because of their lifestyles and morphofunctional convergence upon that seen in modern crocodilians. Semi- or obligate aquatic archosaurs exhibit a range of morphological features not suited for high velocity, bipedal terrestrial locomotion including modified paddle-like limbs, changes in intervertebral joint stiffness (initially lesser but becoming greater as lineages become more aquatic), and reduced limb length relative to trunk length [16–18]. Paddle-shaped limbs are self-evidently less effective at supporting upright body positions. Overly limber or overly stiff vertebral columns do not offer either the stability or flexibility necessary for the maintenance of a horizontal, balanced posture during bipedal movement. Also, reduced limb lengths would be insufficient for achieving the necessary speed or ground clearance.

Figured reconstructions in publications were considered to be indicative of the authors' determination of locomotor mode and of equal merit to textual determination. In instances where only diagrams were presented as the basis for determining the locomotor style, the reconstructed posture of the animal was considered to be indicative of the determination. In instances where both quadrupedal and bipedal diagrams were presented, taxa were determined to be facultative bipeds. We consider this to be justifiable because diagrams only come to exist in the literature as the consequence of a cascade of decisions: firstly, authors have made an intellectual assessment of an animal's posture based on their understanding of the osteological material that is available; secondly, that figure has been produced by the authors themselves or on the authors' behalf (and approved by them); thirdly, the peer-review process has deemed that figure appropriate for publication in a scientific journal. Therefore, the reconstruction must be considered representative of a reasonable scientific understanding of the animal at the time of publication. Taxa with no published locomotor mode were pruned from the dataset because the methods used in this study cannot accommodate unknown character states.

In total, 108 taxa were included in these analyses after pruning, 15 of which were diagnosed on the basis of diagrams alone (electronic supplementary material, file S1). Locomotor mode was treated as a discrete variable with character states 0, 1 or 2 to represent OQ, FB and OB, respectively. We did not impose any directional preference on transitioning from one mode to another, as this would bias the analysis towards finding a certain result and not provide an objective assessment of the evolutionary variability of bipedality. To assess the sensitivity of our analytical approach, we replicated the following analyses using a dataset that excluded 15 taxa for which there was only diagrammatic data available.

The topologies of the two phylogenetic trees were redrawn in Mesquite (version 3.51) [19] and imported into the R statistical environment (version 3.4.3) [20]. Polytomies were randomly resolved into bifurcations using the 'multi2di' function in R package 'ape' [21], as character states cannot be optimized on polytomies using these methods. Random polytomy resolution had no effect on any of the patterns observed in these analyses, because all taxa included in each polytomy were assigned the same locomotor mode. Trees and their branches were dated by first and last appearance in the fossil record using the 'DatePhylo' function in the 'strap' R package [22], using equal share dating. First and last occurrence data were taken from the Paleobiology Database (www.paleobiodb.org).

Ancestral state reconstructions were performed on each tree using the 'ace' function in 'ape' [21]. We opted for a maximum-likelihood ancestral state estimation with discrete character states and an equal rates model of transition rather than a parsimony-based analysis. This reflects the highly variable branch lengths between taxa, whereas parsimony-based analyses assume that each branch of the tree is of equal time length. Maximum-likelihood can account for varied branch lengths by using a rate of evolution algorithm [21]. We used a joint estimation procedure, which incorporates information from all nodes to calculate the maximum-likelihood ancestral state at each node, rather than just the tips and branches descending from that node, as is done in a marginal estimation procedure. This approach gives the most likely combination of ancestral likelihood states [21,23]. Maximum-likelihoods were graphically represented as proportional pie charts at each node in the trees and were plotted using the 'geoscalePhylo' function in 'strap' [22] (figures 1 and 2). Using the most likely character state at each node, we then determined if the acquisition of OB from OQ involved an intermediate FB stage for each independent evolution of OB. The sensitivity analyses were conducted and presented using the same procedures (figures 3 and 4). The R code used in this analysis is available in electronic supplementary material, file S2.

## 3. Results

The Ezcurra tree is focused primarily upon early archosauriforms. In this tree (figure 1), FB is only ever recovered evolving from an OQ ancestor. Likewise, OB is only ever recovered evolving from OQ.

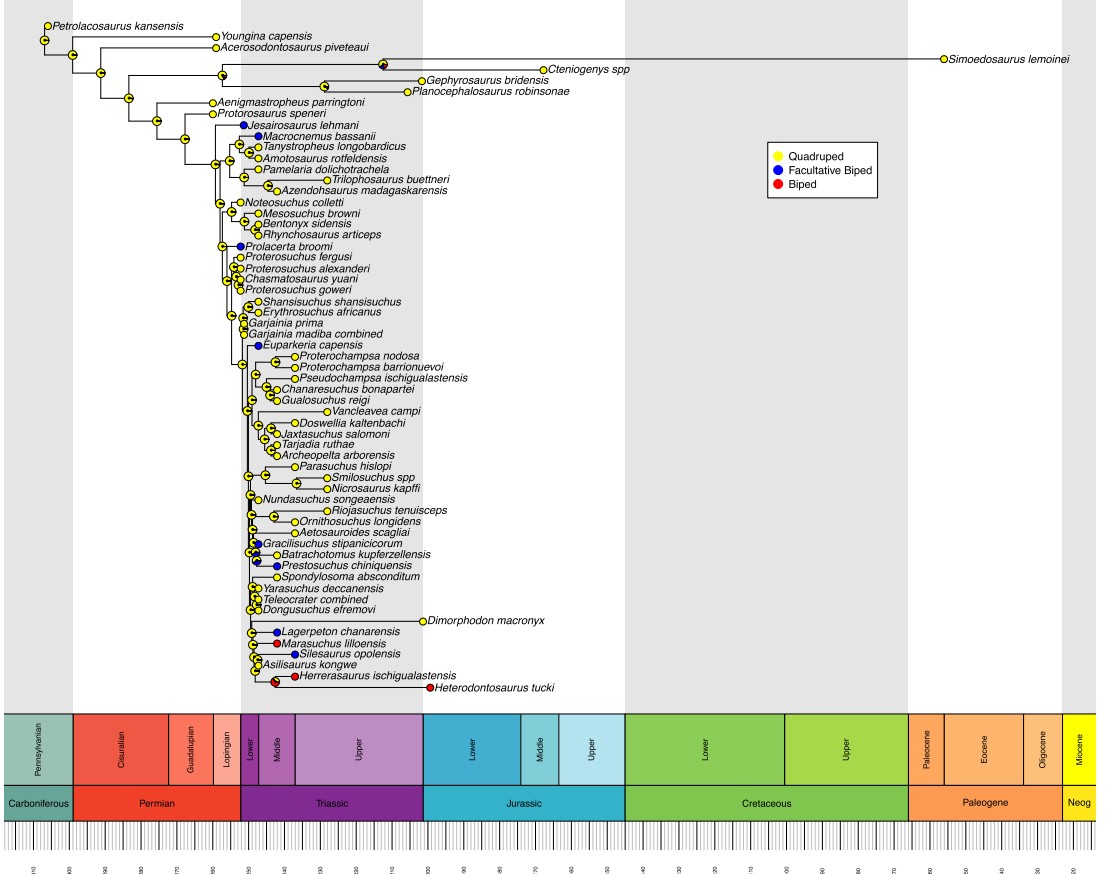

**Figure 1.** Maximum-likelihood reconstruction of ancestral states for locomotor mode based on the time-calibrated Ezcurra tree. Likelihoods are represented by graphical pie charts. Neog = Neogene.

Ancestral states within dinosauriforms remain quadrupedal, despite the end nodes being either facultative or obligate bipeds. Within the Ezcurra tree, we recover eight instances of the independent acquisition of FB, and two instances of OB acquisition. In the sensitivity analysis of the Ezcurra tree, we recover seven independent acquisitions of FB, and two of OB.

The Nesbitt tree (figure 2) includes a wider range of taxa including later archosaurs and early Dinosauria. In this tree, we do recover the expected transition from OQ through FB to OB, but this only occurs once, within Silesauridae. The silesaurid ancestral node (*Asilisaurus* (*Silesaurus* + *Sacisaurus*)) is recovered as quadrupedal, with its descendant node recovered as an FB, and finally, *Sacisaurus* is determined to be an OB. Throughout this tree, there are no other instances of OB emerging from an FB ancestral state, though there are two instances of FB evolving from an OB state. In total, we recover 12 independent acquisitions of FB and seven of OB. In the sensitivity analysis, we no longer recover the OQ–FB–OB sequence because the locomotor mode of *Sacisaurus* was determined on the basis of an anatomical diagram rather than a textual description. Here, we recover a total of 12 independent acquisitions of FB, and five of OB.

Accounting for the overlapping of some taxa across both trees, we recover a total of 14 independent acquisitions of FB and seven of OB across the two trees (figures 1 and 2), reduced to 13 of FB and 5 of OB in the sensitivity analysis (figures 3 and 4). All instances of OB acquisition are found within Archosauria, and the single full transition from OQ through FB to OB occurs in Silesauridae.

Some nodes, such as the ancestor to *Simoedosaurus* and *Cteniogenys* (figures 1 and 3) show OB and FB components despite being deeply located within an OQ dominated section of the tree and having OQ tips. This is an artefact of long branch lengths coupled with the rate of evolution model used in calculating the most likely ancestral state, resulting in likelihoods that entertain the possibility of OB and FB evolving by chance in the ancestor to those species. A similar artefact is also seen in the node ancestral to *Allosaurus* and *Velociraptor* (figures 2 and 4).

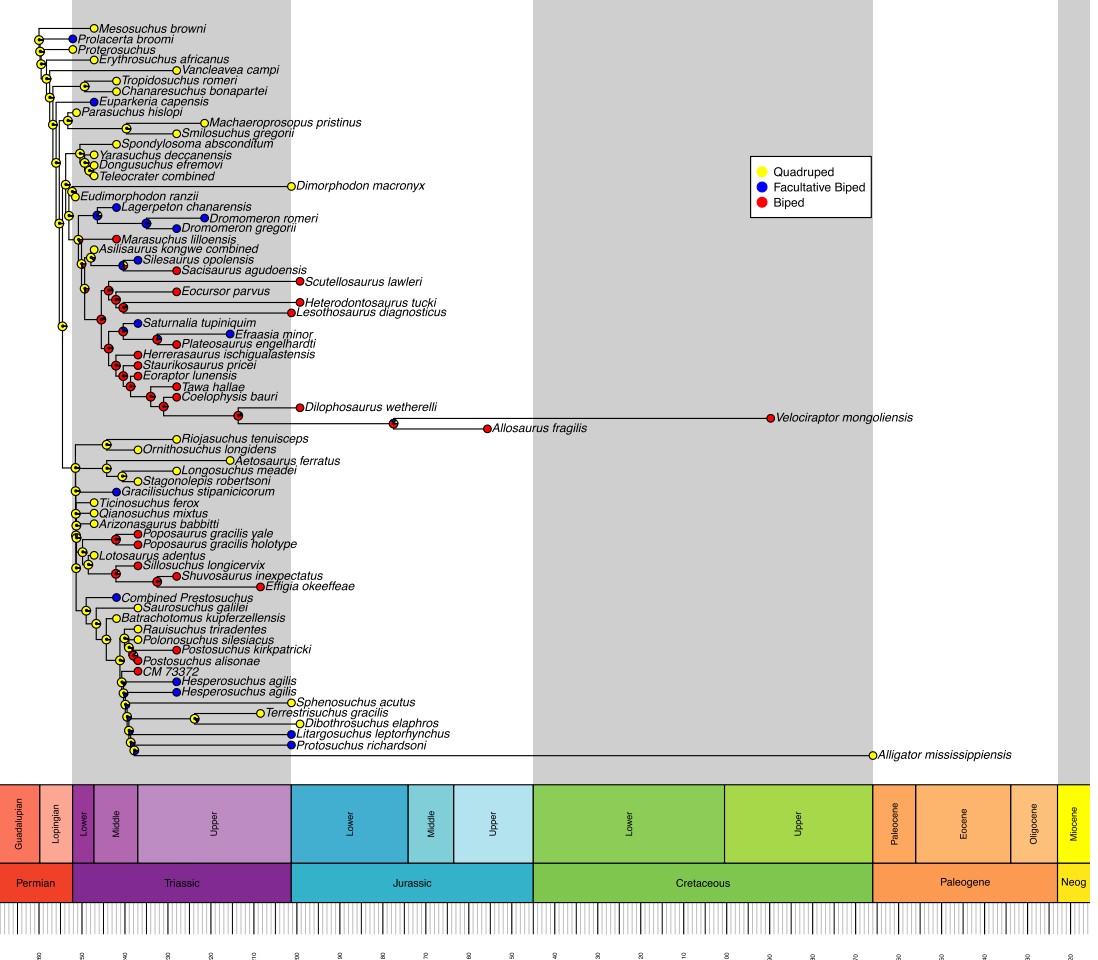

**Figure 2.** Maximum-likelihood reconstruction of ancestral states for locomotor mode based on the time-calibrated Nesbitt tree. Likelihoods are represented by graphical pie charts. Neog = Neogene.

## 4. Discussion

We inferred patterns of locomotor mode evolution across two recent early archosaur phylogenies to test whether FB exists as a consistent transitional locomotor mode between the conditions of OQ and OB. We identify just a single instance in which FB forms an intermediate locomotor mode in the evolution of archosaur bipedality from quadrupedality, out of a total of 14 instances of FB evolution and 7 instances of OB evolution. This single example of the OQ–FB–OB transition occurs within the clade Silesauridae, which has a basal sister-group relationship to Dinosauria and does not therefore contribute directly to the origin or emergence of bipedality within Dinosauria.

In the past, a maximum of six independent acquisitions of archosaur bipedality have been hypothesized. That total figure includes instances determined by the authors to be possible, but unconfirmed [11]. Our finding of seven independent acquisitions of obligate bipedality in archosaurs exceeds all other estimates made to date. Considered alongside the 14 acquisitions of facultative bipedality, it implies that the adoption of particular locomotor modes in these reptiles was far more evolutionarily plastic than previously hypothesized. This is strongly supported by the pervasive distribution of facultative bipedality in the Nesbitt tree, which focuses on a more derived range of archosaurs than the Ezcurra tree. These findings contrast markedly with the classic, and perfectly plausible, hypothesis that facultative bipedality played an important transitional role in archosaur locomotor evolution.

There are two common issues in palaeontological research that may affect our results: phylogenetic topology and morphological proxies for behaviour. Data quality is known to affect the robustness of phylogenetic hypotheses [24], and there may not be enough well-known early archosaurs described to establish robust, stable topologies. In this work, we have used the current understanding of the fossil record to investigate evolutionary transitions between diagnosed locomotor modes (electronic

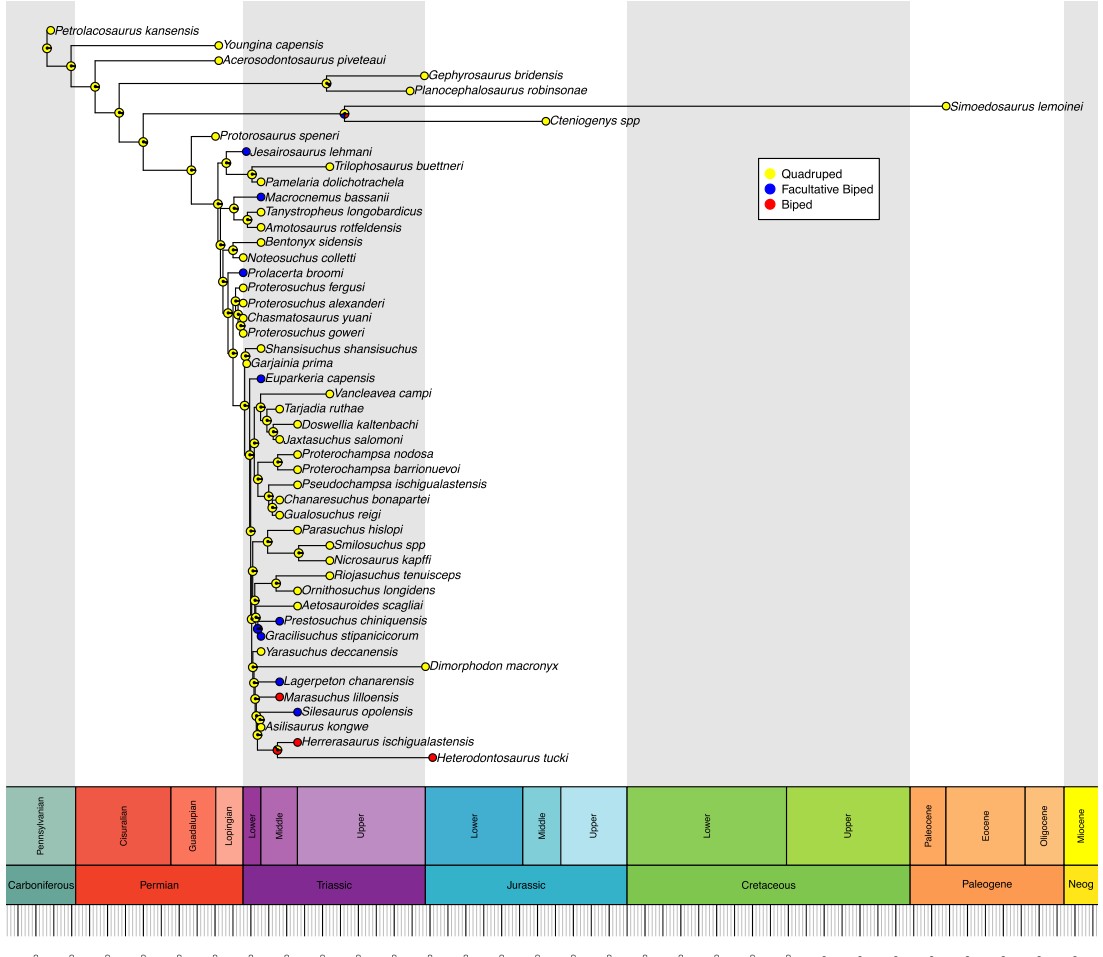

**Figure 3.** Sensitivity analysis, with diagrammatically diagnosed species removed, for the maximum-likelihood reconstruction of ancestral states for locomotor mode based on the time-calibrated Ezcurra tree. Likelihoods are represented by graphical pie charts. Neog = Neogene.

supplementary material, file S1). It is possible that facultatively bipedal taxa not yet described and lying on the dinosauriform stem of Dinosauria will provide support for the existence of a transitional locomotor mode in future. Despite these limitations, advances have been made in studying locomotor transitions in extinct diapsids.

Kubo & Kubo [11] found a significant correlation between their indices for bipedality (humerus plus radius length divided by femur plus tibia length) and cursoriality (metatarsal to femur length) in Triassic archosaurs, suggesting that bipedal archosaurs were also more cursorial, adding a layer of complexity to the evolution of bipedality. Maidment & Barrett [25,26] explored the full scope of traits associated with the evolution of quadrupedal locomotion in Ornithischia. Alongside whole-body traits such as a more cranial centre of mass distribution, this included five readily identifiable osteological correlates relating to muscle attachments or postural shifts. Based on the apparent coevolution of cursoriality and bipedality, and the multifaceted nature of quadrupedal evolution, we infer that the emergence of a bipedal locomotor mode would be similarly mosaic.

It is therefore clear that the identification of locomotor mode in the fossil record has always been challenging, and this uncertainty undoubtedly influences our results. This is especially important when evaluating the potential capacity for a facultative locomotor mode to exist as an intermediary stage in the evolution of bipedality. The means used by different authors to determine locomotor mode have been extremely variable across the history of archosaur research [27–30]. For some species, a robust determination has been made using biomechanical models and in-depth musculoskeletal reconstructions [30,31]. Many studies, particularly older studies, use a deterministic methodology that lacks such a rigorous mechanistic approach and they are thus inconsistent with each other. In some cases, little-to-no justification was given by the author, e.g. relying solely on longer distal limb elements to diagnose facultative bipedality, regardless of other anatomical features [32]. This last

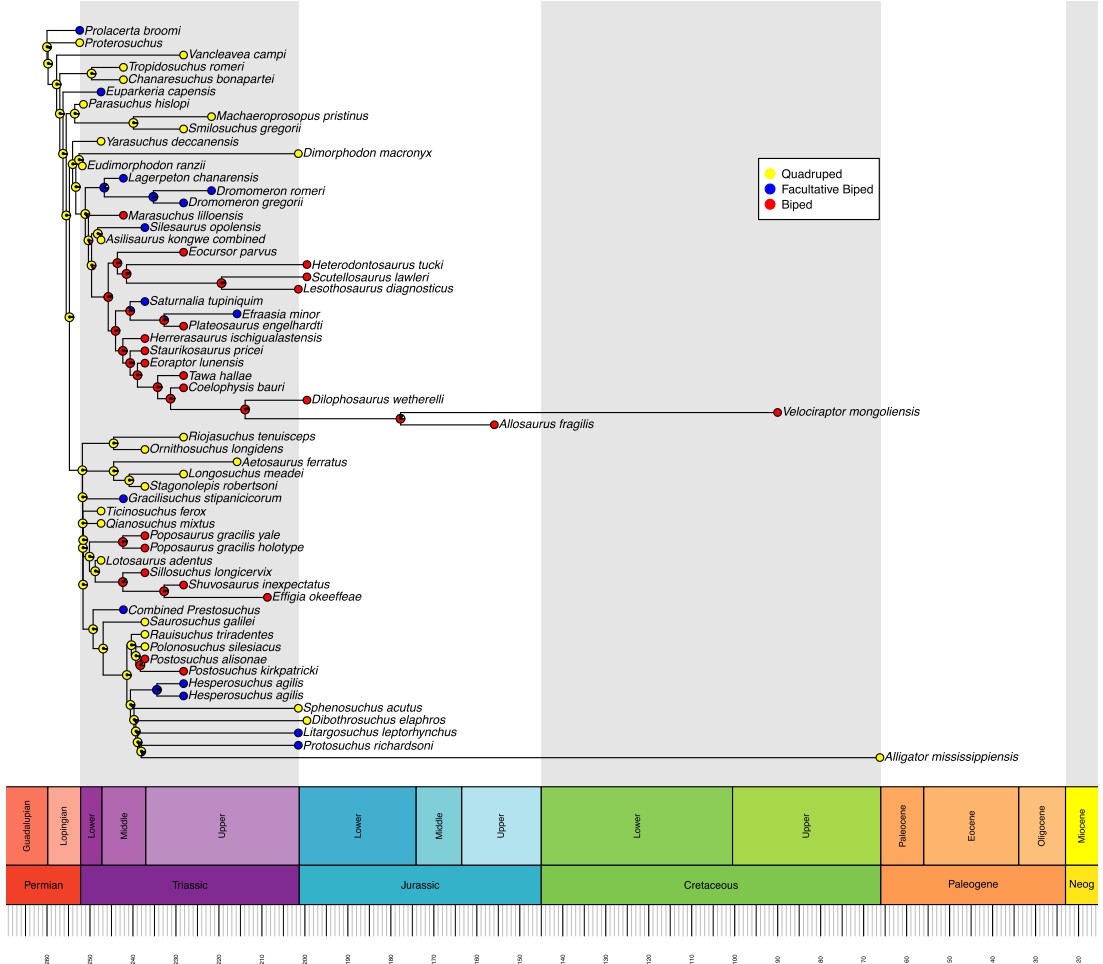

**Figure 4.** Sensitivity analysis, with diagrammatically diagnosed species removed, for the maximum-likelihood reconstruction of ancestral states for locomotor mode based on the time-calibrated Nesbitt tree. Likelihoods are represented by graphical pie charts. Neog = Neogene.

methodological approach, although widely used, stems from data compiled using mammalian limb proportions, rather than a diapsid or multi-taxon dataset; this latter approach commonly relies on forelimb-to-hindlimb ratios as an indicator of bipedality [33]. It should be noted that a musculoskeletal modelling approach does not equate to accuracy, though by the nature of its multidisciplinary methodology it does demand more rigour than inference alone.

When considered in the context of the results presented here, we must entertain the possibility that current interpretations of archosauriform locomotor mode are unlikely to be accurate and under- or mis-identify facultative bipeds in the fossil record. Despite this uncertainty, we do find evidence for FB existing as a transitional mode in this study, as has been hypothesized as widely accepted by the palaeontological research community for some time. However, we do not find evidence for this in the direct ancestors of Dinosauria. Our results find only one example of the predicted evolutionary sequence, which occurred when taxa determined on the basis of anatomical reconstructions alone were included. This result highlights issues regarding the identification of locomotor mode, particularly FB, in the archosaur fossil record. Ultimately, the literature-based determinations of the locomotor mode used in this analysis have been made by experts in their respective fields using their own anatomical knowledge, inference and understanding of the biomechanics of archosaurs. Therefore, the analyses presented here are based on the most current interpretations of the archosaur fossil record.

If it transpires that we are currently identifying facultatively bipedal archosaurs at the correct frequency in the fossil record, further investigation is warranted into the mechanisms of acquiring an obligate bipedal locomotor mode directly from an obligate quadrupedal one. In the light of these results, a systematic review of archosaur locomotor mode is required to more accurately test the hypothesis of FB forming a necessary intermediate mode in the acquisition of dinosaur bipedality. Following a rigorous analysis of traits emerging among the first bipedal archosaurs, in a similar vein to Maidment & Barrett in recent years

[25,26], a thorough analysis of the emergence of these traits across the phylogenies presented here should be conducted. A particular focus of such work should be on the instances of bipedal evolution recovered in the analyses presented here. The primary difficulty that we anticipate in such a programme of work lies in the rarity of good-quality osteological material from exclusively Late Triassic and Early Jurassic locations. The previous worked example [25,26] used what appears to be a better-quality (exclusively ornithischian) fossil record based almost exclusively on large dinosaurs, and had the benefit of spanning a considerably greater time range.

## 5. Concluding remarks

Using the most recent phylogenetic hypotheses and a range of rationales for locomotor mode determination, we recover seven independent evolutionary origins of obligate bipedality among archosaurs. Of these, only a single complete evolutionary transition via a facultative locomotor mode from an obligate quadrupedal one exists, although this result is not recovered in our sensitivity analysis. Our results therefore suggest that facultative bipedality is *not* a necessary transitional locomotor state in the evolution of archosaur bipedality (as hypothesized repeatedly in the past) and does not form an essential stage in the evolution of dinosaurian bipedality. We suggest that this unexpected result might be attributable to inconsistent interpretation of the morphology associated with facultative bipedality in a wide range of archosaur studies, or alternatively that archosaurs are acquiring an obligately bipedal locomotor mode via unexplored anatomical mechanisms.

A renewed assessment and interpretation of the morphological traits associated with locomotor mode, particularly facultative bipedality, in early archosaurs seems to be necessary if we are to more accurately interpret the evolutionary transition to bipedality in this group and properly test the novel hypothesis implicit in this analysis.

Data accessibility. Sources for locomotor diagnoses and applicable evidence are available in electronic supplementary material, file S1. Code for analysis is available in electronic supplementary material, file S2, with Ezcurra and Nesbitt matrices available in their cited work [14,15].

Authors' contributions. L.R.G. conceived of the study, designed the study, collected the data, carried out the statistical analyses and drafted the manuscript; C.S.V.B. conceived of the study, designed the study, helped with statistical analyses and helped draft the manuscript; D.B.N. conceived of the study and helped draft the manuscript. All authors gave final approval for publication.

Competing interests. We have no competing interests.

Funding. Funding for this work came from a NERC DTP grant (1772911) awarded to L.R.G.

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
