## [Reviewer comments · Royal Society Open Science]

Review History

RSOS-182048.R0 (Original submission)

Review form: Reviewer 1 (Andrew Cuff)

Is the manuscript scientifically sound in its present form?

Yes

Are the interpretations and conclusions justified by the results?

Yes

Is the language acceptable?

Yes

Is it clear how to access all supporting data?

Yes

Do you have any ethical concerns with this paper?

No

Have you any concerns about statistical analyses in this paper?

No

Recommendation?

Major revision is needed (please make suggestions in comments)

Comments to the Author(s)

The manuscript "Testing for a facultative locomotor mode in the acquisition of archosaur bipedality" draws upon interpretations of various archosauromorph postures from the literature and tests whether facultative bipedality is a transition phase between quadrupedality and obligate bipedality.

I find the manuscript to be incredibly well written and found no grammatical errors.

The biggest limitation of the manuscript is that it relies on the notoriously difficult assessment of bipedality and quadrupedality in the fossil record as the authors describe in the final paragraph of the discussion. I believe the authors should take this further and put in an additional table (supplementary would be fine), that shows a more extensive version of what is shown in S1 already. A table that contains the references, the posture, and method used in the paper for ascertaining posture (or none/figured only) would then allow readers to make their own judgment, and allow future workers to prune the tree based on different methodologies (and be an incredibly useful resource).

Additionally, in both figures there are some unusual ancestral states for certain nodes that I cannot make sense of. For example, in Figure 1 (Ezcurra tree), the node between Simoedosaurus and Cteniogenys shows just under a half the likelihood being bipedal (both facultative and obligate around 1/4 total) when all the stemward nodes show quadrupedality, as do the two genera. Similarly, in Figure 2, the node between Allosaurus and Velociraptor shows increasing likelihoods of quadrupedality or facultative bipedality when the nodes below are almost exclusively bipedal until dinosauromorphs. I believe these need explaining.

Minor formatting note, check your *Genus species* italics in your references are standardised.

Review form: Reviewer 2

Is the manuscript scientifically sound in its present form?

Yes

Are the interpretations and conclusions justified by the results?

Yes

Is the language acceptable?

Yes

Is it clear how to access all supporting data?

Not Applicable

Do you have any ethical concerns with this paper?

No

Have you any concerns about statistical analyses in this paper?

No

Recommendation?

Accept with minor revision (please list in comments)

Comments to the Author(s)

Please see attached file (Appendix A).

Decision letter (RSOS-182048.R0)

18-Feb-2019

Dear Mr Grinham:

Manuscript ID RSOS-182048 entitled "Testing for a facultative locomotor mode in the acquisition of archosaur bipedality" which you submitted to Royal Society Open Science, has been reviewed. The comments from reviewers are included at the bottom of this letter.

In view of the criticisms of the reviewers, the manuscript has been rejected in its current form. However, a new manuscript may be submitted which takes into consideration these comments.

Please note that resubmitting your manuscript does not guarantee eventual acceptance, and that your resubmission will be subject to peer review before a decision is made.

Your resubmitted manuscript should be submitted by 18-Aug-2019. If you are unable to submit by this date please contact the Editorial Office.

on behalf of Dr Monica Daley (Associate Editor) and Kevin Padian (Subject Editor)
openscience@royalsociety.org

Associate Editor Comments to Author (Dr Monica Daley):

Associate Editor: 1

Comments to the Author:

Your paper has been evaluated by two experts, who both agree that the paper is well written and makes a useful contribution to the literature; however, both reviewers also raise the same concern that the main conclusions of the paper are drawn from interpretations of posture from the literature that can be notoriously difficult and unreliable. The reviewers provide some constructive suggestions for more thoroughly and transparently addressing this issue within the paper.

Reviewers' Comments to Author:

Reviewer: 1

Comments to the Author(s)

The manuscript "Testing for a facultative locomotor mode in the acquisition of archosaur bipedality" draws upon interpretations of various archosauromorph postures from the literature and tests whether facultative bipedality is a transition phase between quadrupedality and obligate bipedality.

I find the manuscript to be incredibly well written and found no grammatical errors.

The biggest limitation of the manuscript is that it relies on the notoriously difficult assessment of bipedality and quadrupedality in the fossil record as the authors describe in the final paragraph of the discussion. I believe the authors should take this further and put in an additional table (supplementary would be fine), that shows a more extensive version of what is shown in S1 already. A table that contains the references, the posture, and method used in the paper for ascertaining posture (or none/figured only) would then allow readers to make their own judgment, and allow future workers to prune the tree based on different methodologies (and be an incredibly useful resource).

Additionally, in both figures there are some unusual ancestral states for certain nodes that I cannot make sense of. For example, in Figure 1 (Ezcurra tree), the node between *Simoedosaurus* and *Cteniogenys* shows just under a half the likelihood being bipedal (both facultative and obligate around 1/4 total) when all the stemward nodes show quadrupedality, as do the two genera. Similarly, in Figure 2, the node between *Allosaurus* and *Velociraptor* shows increasing likelihoods of quadrupedality or facultative bipedality when the nodes below are almost exclusively bipedal until dinosauromorphs. I believe these need explaining.

Minor formatting note, check your *Genus species* italics in your references are standardised.

Reviewer: 2

Comments to the Author(s)

Please see attached file.

Author's Response to Decision Letter for (RSOS-182048.R0)

See Appendix B.

RSOS-190569.R0

Review form: Reviewer 1 (Andrew Cuff)

Is the manuscript scientifically sound in its present form?

Yes

Are the interpretations and conclusions justified by the results?

Yes

Is the language acceptable?

Yes

Is it clear how to access all supporting data?

Yes

Do you have any ethical concerns with this paper?

No

Have you any concerns about statistical analyses in this paper?

No

Recommendation?

Accept with minor revision (please list in comments)

Comments to the Author(s)

I have re-reviewed the manuscript and thing the changes in the supplementary information showing the method of diagnosis/text associated is a good addition as are the sensitivity analyses. I only have a couple of minor changes.

Abstract lines 19-20: You can remove the sentence about the evolution of bipedality being well understood in hominins and not in archosaurs.

Figure 3 and 4 captions: It is worth mentioning what you did in your sensitivity analyses. e.g. Sensitivity analysis where species with just figured diagnoses were removed.

Decision letter (RSOS-190569.R0)

03-Jun-2019

Dear Mr Grinham

On behalf of the Editor, I am pleased to inform you that your Manuscript RSOS-190569 entitled "Testing for a facultative locomotor mode in the acquisition of archosaur bipedality" has been accepted for publication in Royal Society Open Science subject to minor revision in accordance with the referee suggestions. Please find the referees' comments at the end of this email.

The reviewers and Subject Editor have recommended publication, but also suggest some minor revisions to your manuscript. Therefore, I invite you to respond to the comments and revise your manuscript.

- Ethics statement

- Data accessibility

<http://datadryad.org/submit?journalID=RSOS&manu=RSOS-190569>

- Competing interests

- Authors' contributions

- Acknowledgements

- Funding statement

Because the schedule for publication is very tight, it is a condition of publication that you submit the revised version of your manuscript before 12-Jun-2019. Please note that the revision deadline will expire at 00.00am on this date. If you do not think you will be able to meet this date please let me know immediately.

Kind regards,
Royal Society Open Science Editorial Office

on behalf of Dr Monica Daley (Associate Editor) and Kevin Padian (Subject Editor)
openscience@royalsociety.org

==Associate Editor Comments to Author (Dr Monica Daley):

This paper provides a thorough review of existing literature interpretations of archosaur locomotor mode, analyzed within a phylogenetic framework to provide a novel perspective on the acquisition of bipedality. This presents a useful perspective on the inferences that can be made based on the current state of knowledge. Considering that the authors have addressed the comments raised in the previous reviews as far as realistically feasible within the framework of the current approach, I am happy to accept the paper for publication subject to minor revisions to more thoroughly address the remaining concerns.

The primary concern raised in the initial round of reviews centered around the heavy reliance on the reconstructions by others as evidence for locomotor mode, which are based upon varied and often unclear methods and quality of evidence. This limitation largely stands. The additionally sensitivity analysis helps address the most questionable of these sources, but the text-based interpretations of locomotor mode are also based on varied and often unclear evidence. It would have been more informative and useful if the evidence extracted from the literature was consistently based on specific osteological features, with well-stated reasoning and assumptions about the relationships between morphology and function. However, such an undertaking may be impossible based on available data. Despite this limitation, the paper does provide a novel analysis and a thought provoking review of existing interpretations of locomotor mode. However, it is hard to build upon a piece of work when it is unclear what specific osteological evidence has informed the interpretations of locomotor mode.

In the final revisions on this paper, I urge the authors to consider providing a more critical and thorough discussion of the limitations of current reconstructions of locomotor mode, and to take particular care in the interpretations drawn from the current evidence. The interpretations should make clear the uncertain and varied nature of the evidence.

As an example of some text that still seems problematic: On lines 270-273, the authors suggest that limb length proportions are not a robust method of evidence for facultative bipedality, but the analysis presented in this paper is not directly based upon limb length proportions or any other consistent specific line of osteological evidence. The language of the interpretations needs to be worded very carefully to make clear what lines of evidence are being drawn from the current paper, and which statements are based upon specific analyses or data presented elsewhere in the literature.

On line 254: the first sentence of this paragraph focuses on 'facultative bipeds', however some of the issues raised within the paragraph on the challenges and limitations of interpreting locomotor mode seem to apply generally, not just to facultative bipedality.

In addressing limitations of the work, the authors acknowledge the challenge of 'morphological proxies for behavior' but, in fact, the challenges go beyond this. The literature-based inferences of locomotor mode, while made by experts in the field, likely involve a broad range of unstated knowledge, as well as assumptions that may or may not be fully supported by evidence. It would be useful if the authors more clearly and thoroughly addressed this issue within the text.

==Reviewer comments to Author:

Reviewer: 1

Comments to the Author(s)

I have re-reviewed the manuscript and thing the changes in the supplementary information showing the method of diagnosis/text associated is a good addition as are the sensitivity analyses. I only have a couple of minor changes.

Abstract lines 19-20: You can remove the sentence about the evolution of bipedality being well understood in hominins and not in archosaurs.

Figure 3 and 4 captions: It is worth mentioning what you did in your sensitivity analyses. e.g. Sensitivity analysis where species with just figured diagnoses were removed.

Author's Response to Decision Letter for (RSOS-190569.R0)

See Appendix C.

Decision letter (RSOS-190569.R1)

24-Jun-2019

Dear Mr Grinham,

I am pleased to inform you that your manuscript entitled "Testing for a facultative locomotor mode in the acquisition of archosaur bipedality" is now accepted for publication in Royal Society Open Science.

on behalf of Dr Monica Daley (Associate Editor) and Kevin Padian (Subject Editor)
openscience@royalsociety.org

Appendix A

Review – Grinham et al., “Testing for a Facultative Locomotor Mode in the Evolution of Archosaur Bipedality”

It has long been clear that multiple transitions to bipedal posture occurred in the course of archosaur evolution, but how those transitions took place has been a matter of speculation and debate. The present paper sets out to address this question by characterising a wide range of archosaurs and closely related reptiles as obligately quadrupedal (OQ), facultatively bipedal (FB) or obligately bipedal (OB), and using a maximum likelihood analysis to infer ancestral states for locomotor mode across two recent phylogenies containing different (but overlapping) sets of these taxa. The results suggest that more transitions to OB took place in the course of archosaur evolution than has previously been recognised, and also that none of the transitions appears to have involved FB as an intermediate step. These inferences are novel and interesting, with potentially important implications for the evolution of posture and locomotion in archosaurs, but as the authors note they are somewhat compromised by the difficulty of reliably distinguishing among OQ, FB and OB in fossil taxa. The authors base their locomotor mode determinations on assessments published in the literature for individual taxa, but these assessments vary widely in their persuasiveness – some are products of detailed biomechanical analysis, but many others are based on simple heuristic indicators (especially limb proportions) or even basically on intuition. I’m especially suspicious of determinations based solely on “[f]igured reconstructions in publications” (p. 5, line 113), since the postures depicted in the figures seem especially likely to arise from guesswork as opposed to analysis on the part of the original authors. The high level of uncertainty around the locomotor mode determinations arguably undermines the whole analysis presented in the paper, and I certainly wouldn’t place much faith in the results. However, the study is nevertheless useful in showing what can be inferred (at least using a maximum likelihood approach) about the evolutionary history of bipedality in archosaurs based on the current, limited state of knowledge, and in highlighting the need for better assessments of locomotor mode in extinct taxa (as astutely noted in the paper). On this basis, and because the paper is so commendably clear about the limitations of the analysis, I’m happy to recommend publication of the manuscript in *Royal Society Open Science*, subject to some fairly minor revisions.

The reasons for choosing maximum likelihood as the analytical method are explained only briefly (p. 6, lines 134-135), and it would be useful to expand this single sentence for the benefit of readers who (like myself) are not fully up to speed on the nuts and bolts of maximum likelihood algorithms. What exactly is a “joint estimation procedure”? Why do discrete character states and an “equal rates model of transition” make maximum likelihood especially appropriate? Perhaps more importantly, is there any reason why parsimony-based methods of ancestral state reconstruction could not be applied to the data set the authors have assembled? If not, then perhaps repeating the analysis with at least one such method would be a useful exercise to include in the paper. It would be interesting to see whether the results were substantially different from those generated by maximum likelihood, or whether the two methods produced broadly similar inferences regarding the evolution of archosaurian bipedality.

Similarly, it would be worth saying a little more about the decision to treat locomotor mode as an “ordered categorical variable” (p. 6, line 120). I assume this means in practice that transitions directly from OQ to OB or vice-versa were treated as being less likely than transitions between either of these states and FB. If so, the ordering scheme adopted should presumably have favoured recovering postural transitions via FB, so the fact that no such transitions were found despite the ordering scheme being in place seems well worth highlighting in the Discussion. In any case, I would recommend stating explicitly how ordering was handled in the analysis (i.e. what effect the ordering scheme had on the probabilities

of recovering evidence for the various possible types of locomotor transition), given the importance of this issue for the paper.

The manuscript is generally well-written, but does contain some minor grammatical mistakes and poorly worded sentences. I've made a number of editorial suggestions below, but I encourage the authors to make their own careful pass through the manuscript in order to ensure that the text is as correct and readable as possible.

I have some minor comments and suggestions regarding specific passages in the text, as follows:

1. (p. 2, line 20) Consider changing "humans" to "hominins". In the following sentence, it might be worth indicating whether the statement about FB having been hypothesised as an intermediate state is meant to apply only to archosaurs or to both archosaurs and hominins.
2. (p. 2, line 27) Change "instances of bipedality evolution" to "independent acquisitions of bipedality".
3. (p. 3, line 48) Change "freedom conferred upon forelimb for use in functions" to "freeing of the forelimbs for functions".
4. (p. 3, line 51) Add a full stop after the phrase in parentheses.
5. (p. 4, line 65) Delete "an" before "obligate". On the previous line, the phrase "exhibiting some degree of bipedal capacity" is a little vague, and it would be better to mention the characteristic that actually defines facultative bipedality: namely, a tendency to make significant use of both bipedal and quadrupedal locomotion.
6. (p. 4, line 71) Change "dinosauromorphs" to "dinosauromorph".
7. (p. 4, lines 72-73) Change the wording to "However, Kubo and Kubo [12] hypothesised that bipedality arose up to six times in archosaurs more generally..." Later in the sentence, change "to bipedality" to "with bipedality".
8. (p. 4, lines 87-88) Change "approaches in which" to "approaches by which", and consider changing "addressed in future" to "further reduced in the future".
9. (p. 5, line 93) Change "are constructed" to "were constructed".
10. (p. 5, line 95) I assume the consensus trees referred to here are specifically *strict* consensus trees and not some other kind, but it would be worth making this clear. On the next line, "derived" would be a better word than "formed".
11. (p. 5, line 101) Change "topologies represent" to "phylogenies include".
12. (p. 5, line 112) In principle, I don't see why a deep "scull-like" tail would interfere with bipedal locomotion, provided the tail wasn't either unduly massive or insufficiently stiff to act as a counterbalance to the weight of the forequarters. I agree that paddle-like or excessively short limbs are good indicators of non-bipedality, though.
13. (p. 6, line 118) Consider changing "do not deal with" to "cannot accommodate".
14. (p. 6, line 131) Change "www.palaeobiodb.org" to "www.paleobiodb.org".
15. (p. 6, line 135) If I'm understanding the logic of the analysis correctly, "maximum likelihoods" should be changed to something like "relative likelihoods of the different locomotor character states".
16. (p. 7, line 139) Consider changing "encompassed" to "involved".
17. (p. 7, line 143) Consider changing "examined the sequence" to "inferred patterns".
18. (p. 7, line 149) Change "only recovered" to "only ever recovered", and delete the second instance on this line of the word "only".
19. (p. 7, line 150) Looks like "OB ancestor" needs to change to "OQ ancestor".

20. (p. 7, lines 155-156) Change “has a wider scope of species included, ranging to include” to “includes a wider scope of species, such as”.
21. (p. 7, line 160) Change “thus” to “so”.
22. (p. 8, line 167) Consider changing “In the case of an obligately quadrupedal state” to something like “Were an obligately quadrupedal state recovered at this node”.
23. (p. 8, lines 180-181) Consider changing the wording to something like, “...markedly with the classically accepted hypothesis that facultative bipedality played an important transitional role in archosaur locomotor evolution, as is also thought to have been the case in ancestral hominins”.
24. (p. 8, line 184) Change “findings” to “finding”.
25. (p. 8, line 186) Consider changing “readily experimented with” to “evolutionarily labile”, or something similar.
26. (p. 9, line 190) Change “link between their index” to “significant correlation between their indices”, and integrate this sentence with the following one.
27. (p. 9, line 197) Change “musculature traits” to “myological modifications”.
28. (p. 9, line 207) Change “transitionary” to “transitional”. Also, the word “exemplar” seems unnecessary here.
29. (p. 9, line 209) Delete “and interpretation”, since identification of facultative bipeds is really the only relevant issue in the context of your analysis (and “interpretation” is vague anyway).
30. (p. 9, line 211) Change the wording to, “The means used by different authors to diagnose locomotor mode have been extremely variable...”
31. (p. 10, line 212) Change “Some species are diagnosed robustly” to “For some species a robust diagnosis has been made”.
32. (p. 10, line 214) The words “and is thus inconsistent” seem unnecessary here.
33. (p. 10, lines 215-216) Delete the words “a taxon possessing”. On the next line the words “or landmarks” also seem unnecessary.
34. (p. 10, lines 217-221) Aside from “data on mammalian limb proportions”, the idea that greater hindlimb length (relative to forelimb length) should correlate with a greater degree of bipedality draws some support from the simple geometric fact that elongate hindlimbs should (especially in a taxon with erect limb posture) make it easier and more natural to lift the forelimbs clear of the ground. Also, it’s not clear to me why current methods should be biased in favour of underidentifying facultative bipeds, as the text implies. I agree that more systematic methods for determining locomotor mode are needed, but how do we know that current approaches are underidentifying facultative bipeds rather than overidentifying them or, despite the obvious shortcomings, identifying them at about the correct frequency?
35. (p. 10, lines 222-223) It’s hard to tell what exactly is meant by “the methods of acquisition for this locomotor mode”. Should this be changed to something like “how transitions to obligate bipedality occurred in archosaur evolution”?
36. (p. 10, line 232) Consider changing “contradictory” to “unexpected”.
37. (p. 12-13, figure captions) In both captions, expand “ancestral state reconstruction” to something like “reconstruction of ancestral states for locomotor mode”.
38. (Supplementary Information) In the table of sources for information on locomotor mode in various archosaurs, change the heading “Taxa” to “Taxon”, and in the heading of the other column change “Diagnostic” to “Diagnosis”. More importantly, consider adding a third column to indicate the locomotor mode assigned to each taxon following consultation of the literature.

Appendix B

26th March 2019,

The Editor
Royal Society Open Science

Resubmission: **Testing for a facultative locomotor mode in the acquisition of archosaur bipedality.**

Grinham LR, VanBuren, CS, Norman DB

We are pleased to present our resubmission of this article on the evolution of bipedality. We have considered all the remarks made by the reviewers directly and modified the manuscript accordingly. We feel confident that this version is considerably improved in terms of clarity. We address each of the sets of reviewer remarks in sequence below in order to demonstrate that we have taken their commentaries on board in this revised version.

Associate Editor remarks:

Thank you for your overview concerning the general quality of our manuscript. We welcome the supportive commentaries and criticisms from the two 'specialist reviewers'. We have addressed the issue of using unsupported skeletal reconstructions for some taxa (see alterations of the manuscript and substantial improvements made to the Supplementary Table S1). To address this issue, we have also conducted a sensitivity analysis (see Figures 3 & 4) by re-running the analysis after removing taxa for which there is only illustrative material provided by the original authors. The textual changes suggested as well as the discussion points raised by both specialist reviewers have added materially to the quality and clarity of this manuscript and we greatly appreciate their thoughtful input to this process.

The 'subject editor' appears to have misunderstood the context and purpose of this manuscript. It is the case that the subject editor has a particular interest in locomotor posture in ornithomirans because he, contrary to many other views, interprets pterosaurs as bipedal (dinosaur-like) creatures despite contrary indications in their skeletal morphology, soft-tissue and trackway evidence. It is the case that we interpret the three pterosaur taxa featured in these analyses as quadrupeds, based on the most recent published, peer-reviewed works. We respond to his various points below.

Specialist reviewer commentaries:

Reviewer 1.

Paragraph 1. Quality of writing.

We wish that this manuscript had been as well-written as the reviewer claims.

Paragraph 2. Extension of Table S1 regarding taxa assessment.

The focus on the means by which we have attempted to judge quadrupedality vs bipedality in taxa from the fossil record is well made (and echoes our clearly expressed concerns). The recommendation that our supplementary table includes bibliographic references, the posture indicated, and the methodology employed to determine locomotor mode was an excellent one and has been complied with. It has become, as suggested, a useful resource that we hope will be valuable to other researchers.

Paragraph 3. Clarity of results and methodological artefacts.

The 'unusual ancestral states' are an artefact of the long-branches associated with some taxa

and an unavoidable product of the analytical model employed; these artefacts are explained clearly in the revised manuscript.

We would like to thank the reviewer for the enthusiastic support for our article, as well as the perceptive remarks that have introduced clarifications that have improved the quality and accessibility of our article.

Reviewer 2.

We welcome the lengthy introductory paragraphs because they are not only complimentary but demonstrate that the reviewer has engaged with and understood the importance of this contribution and appreciates its overall context.

The comments concerning our reliance upon reconstructed skeleton illustrations is, again, well made and was acknowledged and highlighted in our original article. We have addressed this concern by undertaking sensitivity analyses in which we pruned out the data pertaining to unsupported reconstructions (unsupported by a text-based analysis) and re-ran the analyses (in *both* instances - using the alternate trees) using the residual taxa. The outcomes are fundamentally the same as our original analysis (see Figs 3 & 4) and do not, therefore, detract from our overall conclusions and the cautionary comments that flowed from this work.

We should point out that it is also a gross simplification to assume/imply that all reconstructions are 'fanciful constructs' produced by 'artists' who have no understanding of the underlying anatomy. In the majority of cases the original authors produced their own best attempts to reconstruct skeletal form, body proportions and posture on the basis of detailed descriptive anatomy. In a few instances the reconstructions have been rendered by more skilled artists, but these are also 'informed' by input from the original researchers. It seems unwise to, in effect, 'rubbish' all work of this type because it is actually based upon a detailed appreciation of the underlying anatomy of these taxa; to do so detracts [unintentionally] from the work of many distinguished scientists. Such published images have also been subjected to peer review and found satisfactory, so it should be understood that these were acceptable for a range of reasons at the time of publication.

Despite the understandable caveat the reviewer goes on to express 'happiness' about a recommendation in favour of publication by RSOS.

Likelihood vs Parsimony. This was a comparatively simple choice based on the nature of the data and assumptions that underlie of the analytical protocols: a parsimony-based approach assumes equal branch lengths, whereas maximum likelihood allows for different branch lengths. Maximum likelihood provides the most objective way of assessing time-based distributional data of this kind. We have modified our explanation and re-phrased some ambiguous wording so that the reasoning behind our approach is clearer. The questions apparent in the mind of the reviewer were well founded and have prompted the textual clarifications that have been introduced. In addition, we explain the reasoning behind the joint estimation procedure used, in order to promote clarity as requested by the expert reviewer. This level of methodological clarity is amiss in recent publications using similar methods, and thus we believe attests to the first principles reproducibility of this analysis.

Copy-editing details. We greatly appreciate the time and effort expended by the reviewer in adding a copy-editor-style commentary on our manuscript. The majority of the changes suggested have been complied with (although some are now lost in the substantial re-write). Again, the manuscript has been improved by these changes, and the inevitable discussion between us, that the reviewer's very generous additional work has provoked.

We thank you for your attention and the detailed critiques of our first draft. We trust that this re-draft clarifies matters and addresses all the pertinent issues that were raised.

Yours faithfully,

Luke R. Grinham
Collin S. VanBuren
David B. Norman

Appendix C

10th June 2019,

The Editor
Royal Society Open Science

Revision: **Testing for a facultative locomotor mode in the acquisition of archosaur bipedality.**

Grinham LR, VanBuren, CS, Norman DB

We are pleased to present our revision of this accepted article on the evolution of bipedality. We have considered all the remarks made by the editors and the reviewer directly and modified the manuscript accordingly. We feel confident that this version meets the final requests for clarity.

In instances where clarity was requested, particularly with regards to how literature interpretations can vary, we have addressed these within the body of the manuscript. A single paragraph has been moved and edited to address these problems at a more “natural” time within the manuscript.

We have expanded considerably upon “morphological proxies” and “multiple traits” in more detail, giving significant explanation as to how these have been dealt with in other works, particularly Maidment & Barrett 2012. Additionally, we have addressed the subject editor’s comments regarding future directions by constructing a framework for analysis based upon the Maidment & Barrett study. We also highlight the difficulties that would be faced applying this methodology to early Archosauria.

We have also made the changes suggested by the reviewer.

We hope these changes meet your requests and thank you for the time spent handling this manuscript.

Yours faithfully,

Luke R. Grinham
Collin S. VanBuren
David B. Norman